# Mapping the ultrastructural topology of the corynebacterial cell surface

**Buse Isbilir[1], Anna Yeates[1], Vikram Alva[2], Tanmay A. M. Bharat[1]***

1 Structural Studies Division, MRC Laboratory of Molecular Biology, Cambridge, United Kingdom,
2 Department of Protein Evolution, Max Planck Institute for Biology Tübingen, Tübingen, Germany

* tbharat@mrc-lmb.cam.ac.uk

## Abstract

*Corynebacterium glutamicum* is a diderm bacterium extensively used in the industrial-scale production of amino acids. Corynebacteria belong to the bacterial family *Mycobacteriaceae*, which is characterized by a highly unusual cell envelope with an outer membrane consisting of mycolic acids, called mycomembrane. The mycomembrane is further coated by a surface (S-)layer array in *C. glutamicum*, making this cell envelope highly distinctive. Despite the biotechnological significance of *C. glutamicum* and biomedical significance of mycomembrane-containing pathogens, ultrastructural and molecular details of its distinctive cell envelope remain poorly characterized. To address this, we investigated the cell envelope of *C. glutamicum* using electron cryotomography and cryomicroscopy of focused ion beam-milled single and dividing cells. Our cellular imaging allowed us to map the different components of the cell envelope onto the tomographic density. Our data reveal that *C. glutamicum* has a variable cell envelope, with the S-layer decorating the mycomembrane in a patchy manner. We further isolated and resolved the structure of the S-layer at 3.1 Å-resolution using single particle electron cryomicroscopy. Our structure shows that the S-layer of *C. glutamicum* is composed of a hexagonal array of the PS2 protein, which interacts directly with the mycomembrane via an anchoring segment containing a coiled-coil motif. Bioinformatic analyses revealed that the PS2 S-layer is sparsely yet exclusively present within the *Corynebacterium* genus and absent in other genera of the *Mycobacteriaceae* family, suggesting distinct evolutionary pathways in the development of their cell envelopes. Our structural and cellular data collectively provide a topography of the unusual *C. glutamicum* cell surface, features of which are shared by many pathogenic and microbiome-associated bacteria, as well as by several industrially significant bacterial species.

## Introduction

Cell envelopes serve as primary interfaces between cells and their surroundings. They protect the internal components of the cell [1,2], regulate cell surface

**Data availability statement:** The cryo-EM density map and the cellular tomograms have been deposited to the Electron Microscopy Data Bank (EMDB) under the accession codes EMD-52335 (S-layer cryo-EM density map), EMD-53146 (tomogram of the cell in Fig 1A and S1 Movie) and EMD-53145 (tomogram of the cell in S2 Movie). The atomic model for the *Corynebacterium glutamicum* S-layer structure presented has been deposited to the Protein Data Bank (PDB) with the accession number 9HPN.

**Funding:** This work was supported by the Medical Research Council, as part of United Kingdom Research and Innovation (also known as UK Research and Innovation) [Programme MC_UP_1201/31 to T.A.M.B.]; the Human Frontier Science Program (RGY0074/2021 to T.A.M.B., V.A.); the European Molecular Biology Organization (YIP to T.A.M.B.); the Wellcome Trust (225317/Z/22/Z to T.A.M.B.); the Leverhulme Trust (Philip Leverhulme Prize to T.A.M.B.), and the Lister Institute for Preventative Medicine (Lister Prize to T.A.M.B.). B.I.'s salary was covered by the Human Frontiers Science Program (RGY0074/2021). The funders had no role in study design, data collection and analysis, decision to publish, or preparation of the manuscript.

**Competing interests:** The authors have declared that no competing interests exist.

**Abbreviations:** AG, arabinogalactan; CEMOVIS, cryo-electron microscopy of vitreous sections; cryo-EM, electron cryomicroscopy; cryo-ET, electron cryotomography; F, phenylalanine; FIB, focused ion beam; GL, granular layer; GTDB, Genome Taxonomy Database; IM, inner membrane; IWZ, inner wall zone; K, lysine; LPS, lipopolysaccharide; MM, mycomembrane; MWZ, medial wall zone; OM, outer membrane; OWZ, outer wall zone; P, proline; PG, peptidoglycan; SDS, sodium dodecyl sulphate; SLP, S-layer protein; 2D, two-dimensional; 3D, three-dimensional.

biochemistry [3], and facilitate communication with neighboring cells [4]. In microorganisms, cell envelopes are absolutely critical for survival, facilitating diverse functions such as attachment to surfaces [5,6], biofilm formation [7], and mediating interactions with other species [8]. The composition and architecture of microbial cell envelopes are adapted to the specific environmental conditions that each microbe inhabits, reflecting the evolutionary history of the particular microorganism [9].

Bacterial cell envelopes are complex, multi-layered assemblies that are typically classified into two groups based on their ability to retain Gram stain: Gram-positive bacteria, which are usually monoderm, and Gram-negative bacteria, which are usually diderm [10], with a few notable exceptions [11–14]. In monoderm bacteria, the inner membrane (IM) is covered by a thick peptidoglycan (PG) layer, while diderm bacteria have a thinner PG layer and are enveloped by a second lipid bilayer called the outer membrane (OM), which is often decorated by lipopolysaccharides (LPSs) [15].

Many diderm and monoderm bacteria have a proteinaceous, paracrystalline surface (S)-layer as the outermost coat of their cell envelope [16–24]. S-layers are composed of repeating units of S-layer proteins (SLPs) that form a two-dimensional lattice encapsulating the cell [9,25]. Although maintaining an S-layer requires significant metabolic investment, its preservation throughout evolution in most bacteria and archaea suggests that it provides considerable selective advantages [9]. While some S-layers have not yet been associated with specific functions, many other S-layers play key roles in maintaining cell shape, regulating cellular defence, and promoting pathogenicity [1,2,20].

Corynebacteria are a part of the family *Mycobacteriaceae* (based on the Genome Taxonomy Database [GTDB] [26]) and usually feature an atypical cell envelope that differs both from classical diderm and monoderm bacteria. While classified as Gram-positive due to their staining behavior, corynebacteria are diderm, and their outermost membrane, known as the mycomembrane (MM), is markedly different from typical Gram-negative OMs [14,27]. This distinctive cell envelope contains an IM, a cell wall containing PG and arabinogalactan (AG) layers, together with the MM. The inner leaflet of the MM contains long fatty acyl chains called mycolic acids that are esterified to AG [14], while the outer leaflet of the MM is composed of various lipids including glycolipids, phospholipids and lipoglycans [28]. The list of MM-containing organisms includes several biomedically significant species such as *Corynebacterium diphtheriae*, *Mycobacterium tuberculosis*, and *Mycobacterium leprae*. Previously, direct visualization of the MM was provided by cryo-electron microscopy of vitreous sections (CEMOVIS) and electron cryomicroscopy (cryo-EM) of whole cells, which demonstrated the diderm nature of the corynebacterial cell envelope [29–31]. Corynebacteria include the heavily studied *Corynebacterium glutamicum*, which is extensively used for industrial-scale amino acid production [32]. The *C. glutamicum* MM is further coated with an S-layer made of the PS2 protein [33], which adds another level of complexity to this unusual cell envelope that is incompletely understood at the ultrastructural and molecular levels.

To bridge this gap in our understanding, we have performed characterization of the unusual cell envelope of *C. glutamicum* by using electron cryotomography (cryo-ET) and cryo-EM of focused ion beam (FIB)-milled cells to assign different layers of the cell envelope to cryo-EM density observed in our images. Our data show variable cell surface architecture of *C. glutamicum*, featuring an S-layer with a distinctive repeating pattern decorating the MM in a patchy manner. To further characterize this S-layer, we isolated and solved its structure to 3.1 Å resolution using single particle cryo-EM. Our structure shows that the S-layer is composed of repeating hexameric units of the PS2 protein that form a planar lattice, which partially coats the cells. By combining our S-layer structure with cryo-ET of the cell envelope and bioinformatics analyses, we provide further clues regarding the MM-anchoring mechanisms of the S-layer and offer insights into its conservation and evolution in corynebacteria. Overall, our results offer a detailed molecular-level characterization of the corynebacterial cell surface, bridging past work and laying a strong foundation for future research on this atypical cell surface, which is of significant biotechnological and biomedical importance.

## Results

### The cell envelope of *C. glutamicum*

To examine the cell envelope of *C. glutamicum* in detail, we performed cryo-EM of the *C. glutamicum* strain 541 (ATCC-13058). *C. glutamicum* is a rod-shaped bacterium approximately 0.7 μm in diameter and 1–2.5 μm in length [34]. Its thickness presents a challenge for acquiring high-contrast images of the cell surface using traditional cryo-EM techniques (S1A Fig). To overcome this limitation, we employed FIB-milling to create thin sections of the cells, which allowed us to obtain images with enhanced contrast of the cell envelope. Using this approach, we imaged multiple *C. glutamicum* cells in FIB-milled lamellae, which was facilitated by the tendency of *C. glutamicum* cells to cluster on cryo-EM grids (S1B Fig). We acquired both tilt series of the lamellae to achieve a three-dimensional (3D) understanding of the cell envelope, as well as two-dimensional (2D) projection images to obtain high-contrast images for morphological analysis, specifically to aid comparison with previous work (Fig 1A and 1B, S1C–S1D and S1–S2 Tables) [31]. We observed *C. glutamicum* cells at various stages of the cell cycle, including single cells and dividing cells with visible septa (Figs 1A and S1C). Both in tomograms and in single 2D images, interior details of the cells were discernible, including ribosomes, phosphate granules, and a nucleoid region, in addition to the layers of the cell envelope (Figs 1A–1B, S1C–S1D and S1–S2 Movies).

At the outermost layer of the *C. glutamicum* cell envelope, a patchy, fragmented S-layer was observed (S1A, S1D and S2 Figs). Previous studies have reported that the coverage of the S-layer on *C. glutamicum* cells depends on the carbon source in the medium and the strain used [35,36]. Consistent with this reported variability [35], we found that cells cultured on agar plates had higher S-layer coverage on the MM when compared to cells grown in liquid media (S2 Fig). In all cases, we observed partially coated *C. glutamicum* cells, with many detached S-layers present around the cell. Beneath the S-layer on the cell, a high-contrast MM layer was observed (Fig 1C and 1D). In 2D projection images of FIB-milled cells, the two leaflets of the MM were resolved (Fig 1C and 1D). The thickness of the MM in cell envelopes with and without S-layer was 7.1 nm and 6.9 nm, respectively (S1 Table). In the case of cells without an S-layer, i.e., those containing a 'naked' MM, the MM presents itself as two continuous, uninterrupted, apposing leaflets, which are clearly separated from each other (Fig 1C). This contrasts with S-layer-coated cell envelopes, where the two lines of density corresponding to the MM leaflets are interrupted and discontinuous (Fig 1D). Although we cannot be certain given the existing data, we suppose that this perturbation of the MM directly beneath the patchy S-layer could arise from the interaction of the S-layer anchoring segment with the MM, which has been predicted to be present at the C-terminal end of the SLP PS2 [9,33].

Below the MM, there is a layer with comparatively weak density that we name the outer wall zone (OWZ), which is interrupted by densities emanating from the MM (Fig 1C and 1D). This layer was not resolved in previous studies, and we propose that this comprises the AG layer, which is covalently connected both to the MM and PG [14]. Moving inward from the OWZ, there is an electron-dense layer called the medial wall zone (MWZ), which was also observed previously [31],

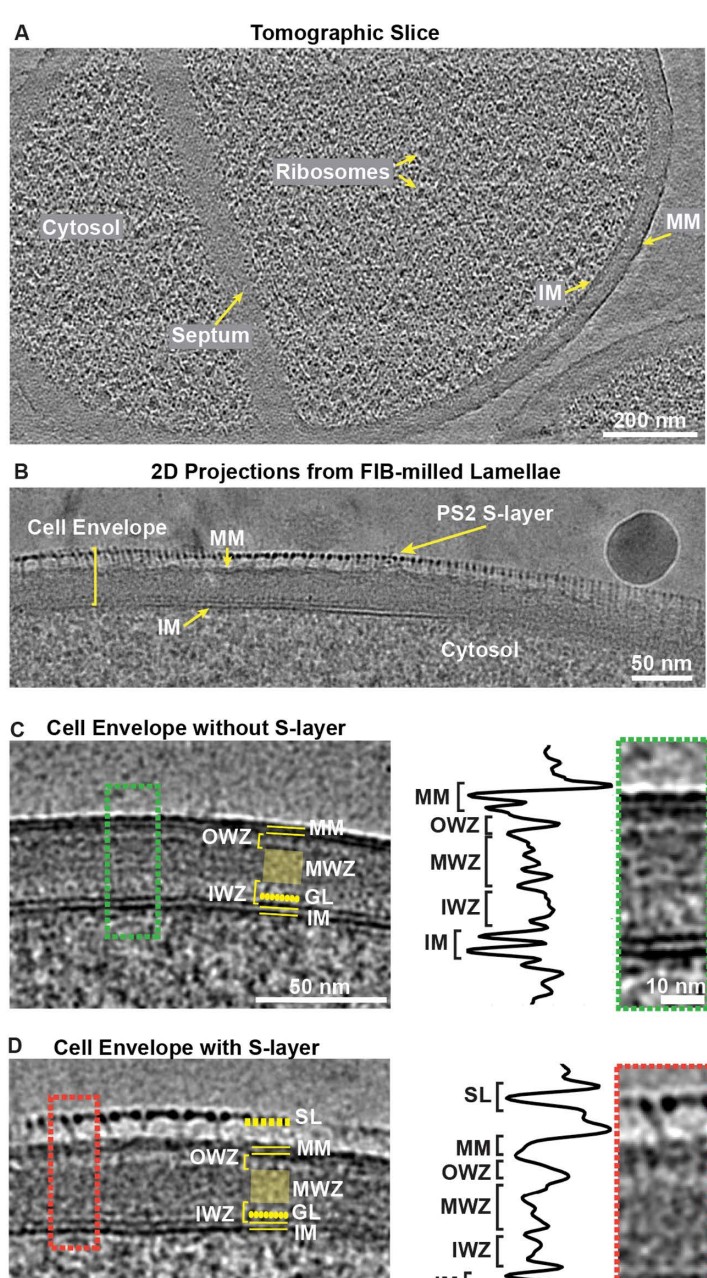

**Fig 1. Ultrastructural investigations of the *Corynebacterium glutamicum* cell envelope. (A)** A slice from a tomogram of a FIB-milled, dividing *C. glutamicum* cell (S1 Movie). Ten 0.85-nm thick slices were averaged and bandpass-filtered to boost contrast. **(B)** Cryo-EM image of a FIB-milled *C. glutamicum* cell. The S-layer decorates the cell in a patchy manner (see also S1 Fig). **(C)** 2D images of the FIB-milled cells without or **(D)** with S-layers. The labeled features are: SL (surface layer), MM (mycomembrane), OWZ (outer wall zone), MWZ (medial wall zone), IWZ (inner wall zone), GL (granular layer), and IM (inner membrane). Line profiles were plotted from the cropped 2D projection images shown in the green (naked envelopes) and red boxes (S-layer coated envelopes). The images were Gaussian-filtered to enhance contrast.

and is predicted to constitute the PG. We found that this PG layer in cell envelopes with and without S-layer was 13.4 nm and 14.2 nm thick, respectively, with an amorphous appearance (Fig 1C–1D and S1 Table). Below the MWZ, we observed another weak layer called the inner wall zone (IWZ), which has been proposed to resemble the periplasmic space of Gram-negative bacteria [31]. The IWZ is partly occupied by densities protruding from the IM, and this region is called the granular zone or granular layer (GL), located ~3.8 nm from the edge of the IM. The presence of the GL is a property of Gram-positive, monoderm bacterial PG and is thought to contain lipoproteins [14]. Both the inner and outer leaflet of the IM are resolved in our data, with the thickness of the IM measured in cell envelopes with and without S-layer as 7.2 nm and 7.8 nm, respectively. Overall, we visualized all the postulated layers of the *C. glutamicum* cell envelope and found no significant difference between naked cell envelopes and S-layer-coated envelopes, apart from interruptions in the MM directly below the S-layer (Fig 1C and 1D and S1 Table).

To probe the plasticity of the cell envelope during the cell cycle, we analyzed the cell envelope layers within the dividing septum (S1E Fig). The thickness of the septum (~55 nm) was found to be greater than the usual thickness of the cell envelope (~42 nm on the same cell, see also Fig 1A). The septum is composed of unseparated cell envelopes of the daughter cells that appear to contain a single 'outer' membrane, which is likely composed of mycolic acids. Presumably, this membrane will form the future MM once division is completed. Notably, the putative mycolic acid-containing bilayer within the septum was not connected to the MM on the other parts of the cell, whereas the remaining cell envelope layers appeared to be continuous with the rest of the cell. While IM and the putative future MM were clearly distinguishable, PG and AG could not be differentially identified in the dividing septum.

## Cryo-EM analysis of the *C. glutamicum* PS2 S-layer

To understand how the S-layer is arranged on *C. glutamicum* cells, we sought to determine the atomic structure of the PS2 S-layer. To this end, we purified the S-layer from the *C. glutamicum* cell surface by adapting previous protocols that used sodium dodecyl sulphate (SDS) to detach the S-layer from bacterial cells [9,16,37]. Purified S-layers were deposited on cryo-EM grids and vitrified using methods previously described for other S-layers [3,16], and specifically for the *C. glutamicum* S-layer concurrently with this study [9,38]. Imaging the cryo-EM grids of purified S-layers confirmed a hexagonally arranged lattice, as previously predicted (Fig 2A and S3 Table) [9]. Tilted and side views of the S-layer were incorporated to reduce resolution anisotropy and enhance Fourier space coverage [39,40], using high-throughput data collection with a 30° tilt of the specimen stage (see Materials and methods). The resulting data was used to obtain a cryo-EM reconstruction at 3.1 Å global resolution (Figs 2B and S3A–S3B), which allowed us to build an atomic model of the *C. glutamicum* PS2 S-layer (Figs 2C, S3C and S3 Table).

Our atomic model of the S-layer lattice shows that the S-layer is formed solely by the PS2 protein, with no additional density observed for other macromolecules or post-translational modifications. This is consistent with previous results [41]. The PS2 protein is primarily composed of α-helices (with some loop residues), which is unusual in prokaryotic SLPs structurally characterized thus far (Figs 2D, S3C and S3 Movie) [3,16,18,19,22,42], except for the SLP of *Clostridioides difficile* [17]. Amino acid residues 33–445 could be reliably modeled into the cryo-EM map, which was sufficient to visualize the S-layer lattice contacts (Fig 2C and 2D and S3D). The missing N-terminal residues of PS2 (residues 1–32) are predicted to form a signal peptide for Sec-dependent secretion and are presumably cleaved off and not present in the mature SLP that forms the S-layer lattice. Indeed, between residues 30 and 31, there is a predicted cleavage site for signal peptidase I, which cleaves the translocated preproteins from the membrane upon their translocation [43]. At the other end of the PS2 protein sequence, the missing C-terminal residues (446–498) were previously shown to be important for cell envelope anchoring [33] and are also predicted by bioinformatic analyses to form an anchoring segment on cells [9]. Due to detergent (SDS) treatment during S-layer purification, this anchoring segment was likely detached from the MM and remained too flexible to be resolved in our cryo-EM analysis. Instead, weak and disordered density was observed in the cryo-EM map for this part of the PS2 protein (Figs 2D and S3E).

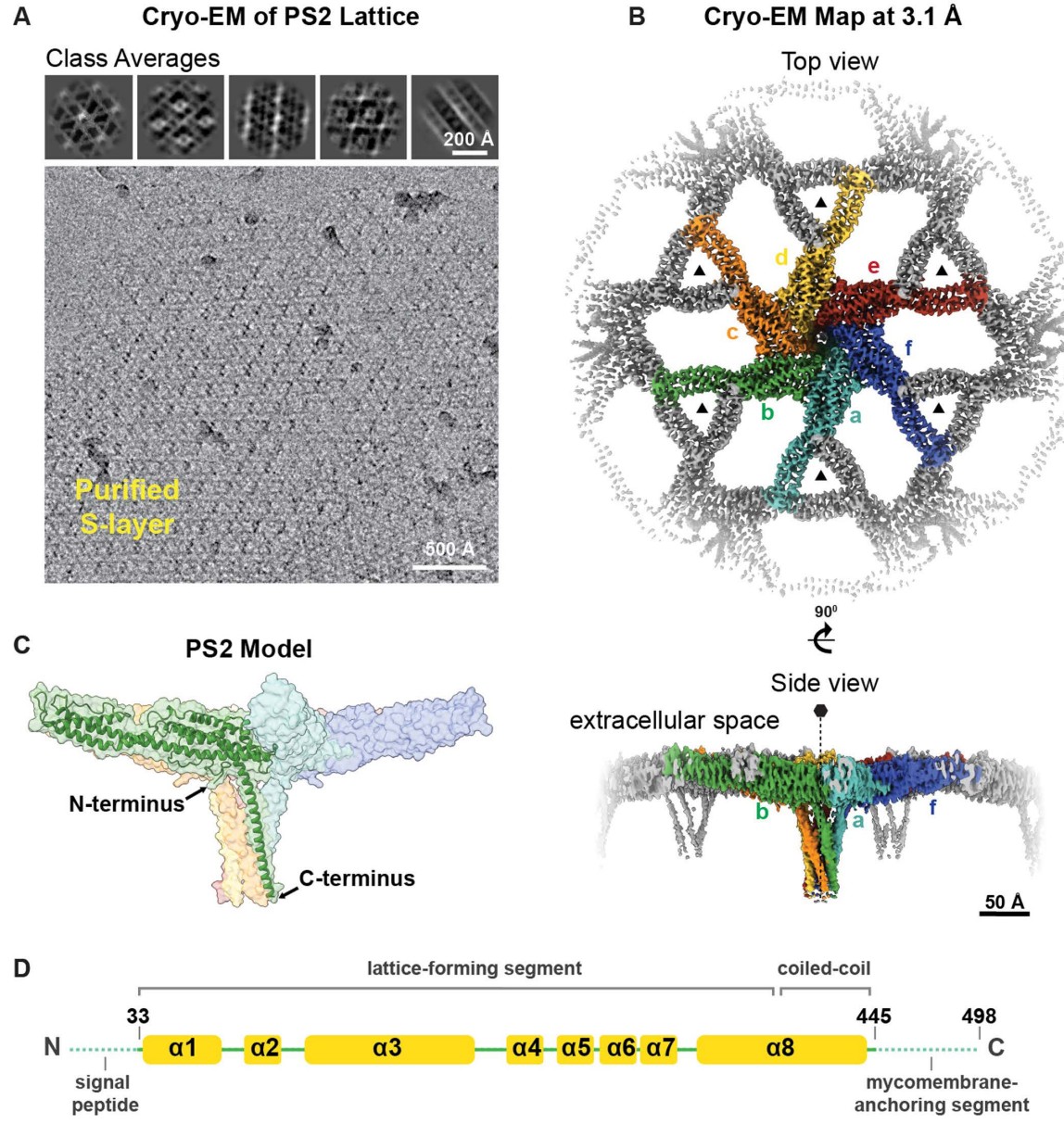

**Fig 2. Cryo-EM structure of the PS2 S-layer of *Corynebacterium glutamicum*. (A)** A cryo-EM image of the PS2 S-layer is shown, together with representative 2D classes (top). The hexagonal pattern of the PS2 S-layer is visible in the images. **(B)** Cryo-EM map of the PS2 S-layer at 3.1 Å global resolution (see also S3 Fig). Top and side views of the map are shown at a contour level 5.6 σ away from the mean value. Each PS2 monomer is colored separately in the hexameric unit, while the rest of the volume is colored gray. **(C)** The PS2 atomic model. The N-terminal part of the protein is involved in lattice formation, while the C-terminal part of the protein forms a coiled-coil segment that protrudes from the lattice to anchor the S-layer to the MM. **(D)** A schematic cartoon of the secondary structure elements in PS2, which is composed of α-helices denoted as α1–α8. Residues 33–445 are built into the density. PS2 amino acid residues that were not built into the density are shown with a dashed line. These unmodeled residues correspond to the N-terminal region of PS2, which carries a signal peptide for its translocation, and the C-terminal region, which contains the MM-binding residues.

## The hexagonal PS2 array on the MM

The extended lattice of PS2 is formed by interactions between the PS2 monomers, mediated by two major symmetry-related interfaces (Fig 3A). The PS2 monomers assemble into tight hexamers within the lattice, which in turn are linked to

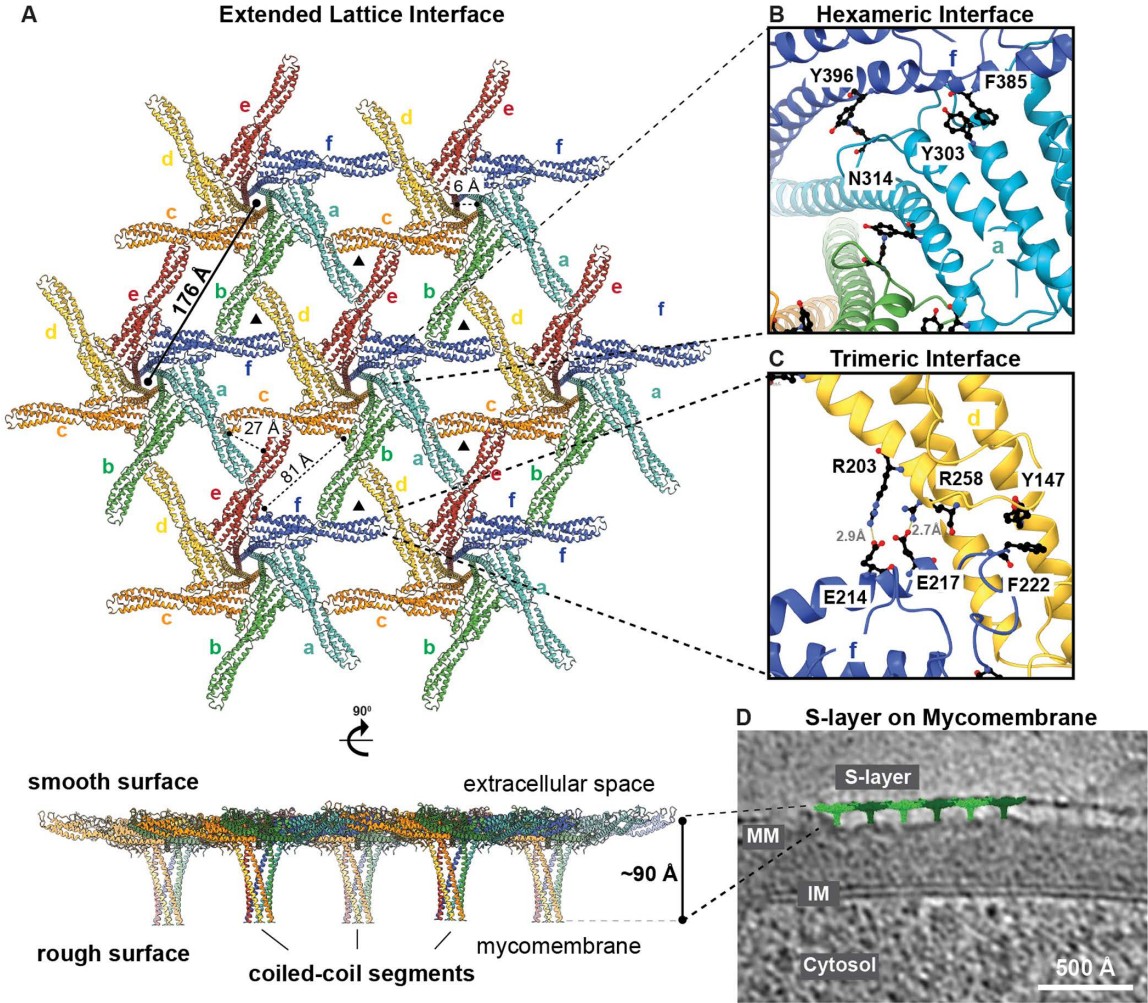

**Fig 3. The PS2 S-layer lattice. (A)** The atomic model of the PS2 lattice is shown in top and side views. PS2 hexamers are repeated in the 2D sheet with a lattice constant of 176 Å. Each PS2 monomer is colored and labeled separately within the hexamer, which repeats throughout the lattice (with pore dimensions highlighted). The side view of the lattice shows the smooth surface facing the extracellular space and rough surface with coiled-coil segments protruding towards the MM. **(B, C)** Interactions at the (B) hexameric and (C) trimeric interfaces are stabilized by both hydrophilic and hydrophobic residues. **(D)** A cell envelope tomogram (see also Fig 1) was used to overlay the cryo-EM structure of the S-layer onto the in situ S-layer tomographic density. A single slice of the tomogram is shown, with the overlayed PS2 S-layer lattice in green and the tomogram in grayscale. The S-layer (surface layer), MM (mycomembrane), IM (inner membrane), and Cytosol are labeled.

each other through prominent trimeric interfaces that are repeated throughout the lattice (Figs 3A and S4A). The hexameric linkages are primarily established by residues 90–110 and 300–445, located around the center of the hexamer and in the coiled-coil MM-anchoring segment. Each hexameric unit is connected to neighboring hexamers through trimeric interfaces, mainly involving residues 194–204 and 213–235 (Fig 3B and 3C). Both the trimeric and hexameric interfaces are formed by a combination of hydrophilic and hydrophobic interactions, including salt bridges and π-stacking of aromatic side chains (Fig 3B and 3C).

At the sequence level, the PS2 protein is enriched in acidic amino acid residues, giving it an overall negative charge, with an estimated isoelectric point of 4.25 (S4B Fig). Consistent with this overall negative charge, we observed putative cationic densities at various locations along the PS2 sequence in the cryo-EM map, which are surrounded and stabilized

by negatively charged amino acid residues (S4D–S4F Fig). The identity of these cations cannot be ascertained at the current resolution of our cryo-EM map; however, previous studies on other bacterial S-layers suggest that they may correspond to calcium [18,21,44]. These cations may further stabilize the lattice, similar to other S-layers where cations were found to be essential for lattice formation [18,21,42,44]. To probe this further, we incubated purified PS2 S-layers with either 10 mM EDTA or 10 mM EGTA and examined their effect on the treated S-layers. Following the chemical treatment, S-layer lattices were still intact, with no observable differences under both conditions (S4I Fig). This suggests that these putative cations either do not play a major role in stabilizing the PS2 S-layer or are not accessible for chelation by EDTA or EGTA under the chosen experimental conditions.

At the overall S-layer level, each PS2 protein monomer interacts with four other monomers to form the intricate lattice. Specifically, each monomer interacts with two other monomers via the hexameric interface and two others via the trimeric interface, creating a highly interconnected arrangement (Fig 3A–3C). The lattice is formed by repetition of hexameric units at a distance of 176 Å (Fig 3A), which closely matches the lattice constant previously predicted for this S-layer [9]. The combination of hexameric and trimeric interfaces results in varying pores sizes of 6, 27, and 81 Å within the lattice (Fig 3A). Some of these pores are relatively large and are reminiscent of the porous S-layer of *Deinococcus radiodurans*, which is also patchy on the cell surface [16]. This indicates that the *C. glutamicum* S-layer likely does not function as a molecular sieve, i.e., it has no protective role due to large pore dimensions and patchy cellular coating of the S-layer.

Moving from the plane of the lattice toward the MM, the anchoring segment of PS2 consists of canonical, parallel, hexameric coiled-coils that extend inward from the S-layer lattice toward the cell. This arrangement results in a smooth extracellular side of the S-layer, while the side facing the MM is rough, containing "spikes" formed by the hexameric coiled-coils (Fig 3A and 3D). When viewed from the side, the purified S-layer has a thickness of 91 Å, consistent with the thickness of ~98 Å observed in situ in FIB-milled data from the *C. glutamicum* cell envelope (Figs 1, 3A, 3D and S1 Table). The atomic model of the S-layer lattice, when overlaid with a slice through a cell envelope tomogram of *C. glutamicum*, shows a good agreement with the cellular data, showing a tight match to the lattice thickness and the anchoring coiled-coil densities (Fig 3D), demonstrating that the SDS treatment did not denature the PS2 protein or disrupt the S-layer. Our overlay further shows that the tip of the coiled-coil anchoring segment, resolved in our cryo-EM map and included in the atomic model, ends just before touching the MM. This observation is consistent with the disordered C-terminal residues missing from our density (S3E and S4H Figs), strongly suggesting that the C-terminal residues directly interact with the MM. A slight wavy pattern is observed on the MM at the bases of the PS2 coiled-coil segments (shown in Figs 1 and 3D), appearing as interruptions in the outer leaflet of the MM. This further supports the idea that the anchoring segments intimately interact with the MM, although this observation needs experimental validation at the molecular level.

## Conservation of the PS2 protein across corynebacterial cell surfaces

After characterizing the molecular structure of the PS2 S-layer, we investigated its presence in various species within the *Mycobacteriaceae* family. To this end, we searched for homologs of the *C. glutamicum* PS2 protein in the non-redundant protein sequence database (nr) at NCBI using BLAST [45]. This search yielded a total of 102 matches, almost all of which were PS2 homologs in other species of the genus *Corynebacterium* (S3 Data). Next, to identify highly divergent homologs of PS2 that may have remained undetected by BLAST, particularly in species outside the genus *Corynebacterium*, we conducted sensitive profile hidden Markov model-based sequence searches against representative bacterial proteomes using HHsearch [46,47] and structural searches against the AlphaFold/UniProt50 database using Foldseek [48]. These searches also failed to identify homologs of PS2 outside the *Corynebacterium* genus and further suggested that the α-helical bundle fold exhibited by PS2 is novel, strongly indicating that PS2 originated within the *Corynebacterium* genus. In fact, to date, within the MM-possessing *Mycobacteriaceae* family, S-layers have only been experimentally observed

in some strains of *C. glutamicum* [49] and *Mycobacterium tuberculosis* variant *bovis* bacille Calmette-Guérin (BCG) [50]. While *M. bovis* BCG strains have been reported to exhibit an oblique S-layer arrangement, the identity of the protein(s) forming this structure has not yet been determined.

Having established that the PS2 protein (and *ps2* gene) is distinct to the *Corynebacterium* genus, we next examined its prevalence within this group. To achieve this, we searched for PS2 in a total of 2,256 proteomes from *Corynebacterium* species and their various isolates, including 73 *C. glutamicum* isolates, using BLAST against the RefSeq Genomes database. We identified PS2 in 115 *Corynebacterium* isolates, predominantly in opportunistic pathogens isolated from human clinical and microbiome contexts, such as *C. aquatimens*, *C. aurimucosum*, *C. auris*, *C. minutissimum*, *C. singulare*, and *C. tuberculostearicum* (S3 Data). PS2 was also found in species from environmental and biotechnological contexts, such as *C. occultum*, *C. halotolerans*, and *C. humireducens*. Interestingly, some species, such as *C. glaucum*, *C. felinum*, and *C. minutissimum*, consistently possess PS2 across all isolates. In contrast, others, including the diphtheria toxin-secreting primary pathogens *C. diphtheriae*, *C. ulcerans*, and *C. pseudotuberculosis*, as well as their closely related species *C. rouxii*, *C. belfantii*, and *C. silvaticum*, never harbor PS2. In many other species, such as *C. glutamicum*, *C. aurimucosum*, and *C. aquatimens*, the presence of PS2 varies among isolates. For instance, of the 73 *C. glutamicum* isolates in our RefSeq dataset, 47 possess PS2, suggesting that the PS2 S-layer likely plays beneficial roles in *C. glutamicum* in certain conditions.

To investigate the evolutionary distribution of PS2, we constructed a phylogenetic tree of 1,585 *Corynebacterium* species (Fig 4A) using the de novo workflow (de_novo_wf) of the Genome Taxonomy Database Toolkit (GTDB-Tk) [51]. This method infers evolutionary relationships based on concatenated multiple sequence alignments of 120 conserved bacterial marker genes, providing a robust phylogenetic framework. The resulting tree reveals a sparse yet widespread distribution of PS2, suggesting that the common ancestor of *Corynebacterium* likely possessed PS2 and, by extension, was coated with an S-layer assembled from it (Fig 4A). This aligns with our earlier findings that PS2 is restricted to *Corynebacterium* and likely emerged within this lineage. The patchy distribution across species suggests differential retention, potentially due to varying selective pressures or functional requirements. Consistent with this pattern, the tree also shows a clear separation between PS2-lacking, diphtheria toxin-associated primary pathogens and PS2-containing environmental and predominantly opportunistic pathogens (Fig 4A). Additionally, we constructed a separate phylogenetic tree of 73 *C. glutamicum* isolates using the same approach (Fig 4B). This tree shows a clear divergence between PS2-containing and PS2-lacking isolates (Fig 4B). Many closely related PS2-lacking isolates cluster together within a single clade, whereas PS2-containing isolates are distributed across multiple clades. We compared the genomes of two closely related isolates—the PS2-lacking isolate CS176 and the PS2-containing isolate ATCC 31831—to identify further genetic differences related to the presence or absence of PS2. Our analysis revealed that CS176 lacks not only the gene encoding PS2 but also several adjacent genes both upstream and downstream of *ps2*. Additionally, both strains possess more than 200 distinct genes each. However, we could not establish a direct connection between these genomic differences and the presence or absence of an S-layer.

At the protein sequence level, the length of PS2 from different *C. glutamicum* strains ranges from 490 to 510 amino acid residues, while the pairwise sequence identity between them ranges from 70% to 100% (S5 Fig). Variations are primarily localized to secondary structural elements involved in the formation of trimeric and hexameric interfaces in the S-layer lattice. In fact, a previous study using AFM imaging of S-layer lattices from different strains with varying PS2 sequences revealed differences in unit cell dimensions [49]. Beyond *C. glutamicum* strains, PS2 exhibits much greater variability, with lengths ranging from 452 to 665 residues and pairwise sequence identity ranging from 30% to 80% (S6 Fig). Despite this variability, all PS2 homologs share common sequence features, including a predicted Sec/SPI signal peptide, a coiled-coil segment, a C-terminal MM-binding hydrophobic segment, and an intrinsically disordered region between the coiled-coil and MM-binding segments. Furthermore, AlphaFold2 models of PS2 from different species reveal a high conservation of secondary structural elements, albeit with length variations, and a similar overall 3D fold (S7 Fig).

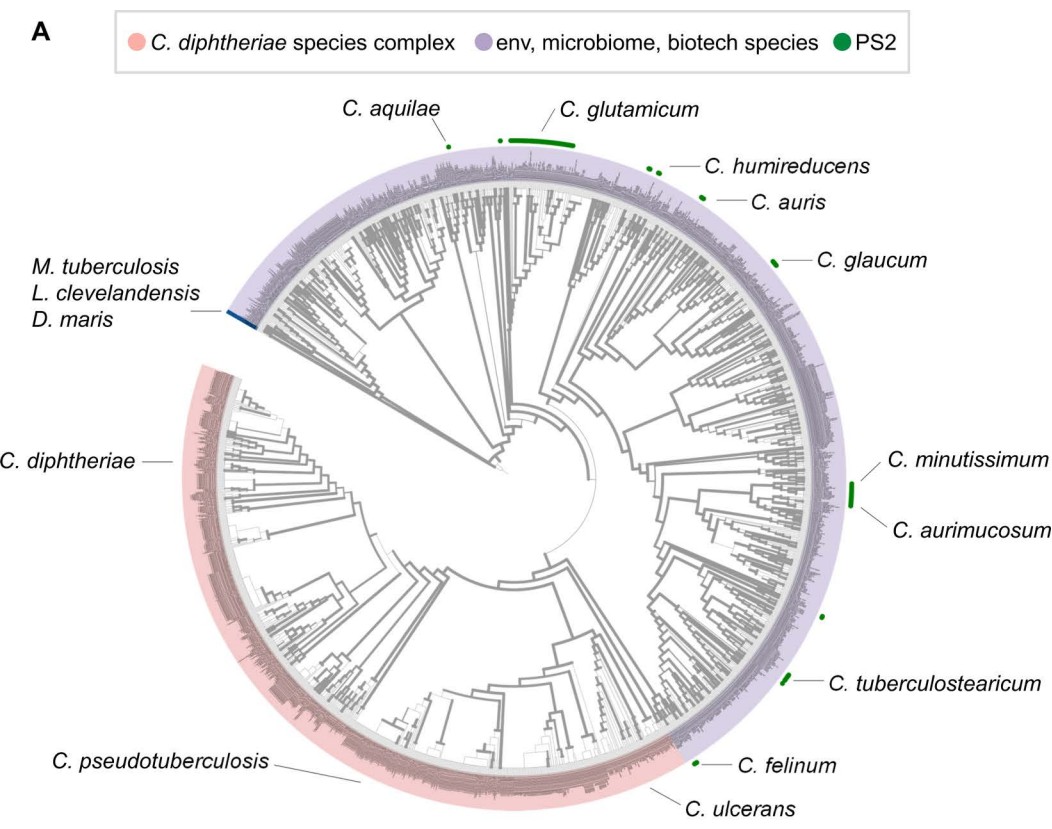

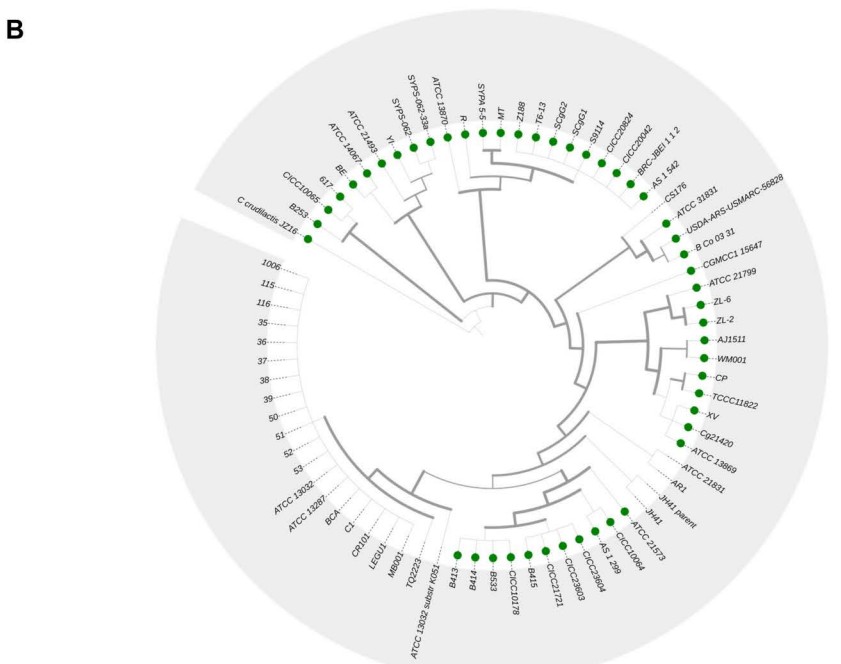

**Fig 4. Phylogenetic distribution of PS2. (A)** The presence of PS2 in species across the *Corynebacterium* species is indicated by green dots on a GTDB (Genome Taxonomy Database, [26]) species-level tree. The outer ring distinguishes diphtheria toxin-associated species (colored light red),

environmental, microbiome-associated, and opportunistic pathogenic species (light purple), and outgroup MM-containing species used to root the tree (blue; *Mycobacterium tuberculosis*, *Lawsonella clevelandensis*, and *Dietzia maris*). Some representative PS2-containing species are labeled. Line thickness represents FastTree2 Shimodaira–Hasegawa-like support values, with thicker lines indicating strong support for the branch and thinner lines suggesting lower branch reliability. The data underlying this tree can be found in S4 Data. **(B)** The presence of PS2 in *C. glutamicum* isolates is indicated by green dots. The tree was rooted using *C. crudailactius* as the outgroup species. The data underlying this tree can be found in S5 Data.

This suggests that all PS2 homologs form S-layers with a similar hexagonal lattice as described in this study, but with lattice parameters that likely vary substantially.

Examining variations in the PS2 amino acid sequence provides valuable insights into key aspects of the corynebacterial cell surface. Given that different corynebacterial species inhabit diverse environments, the varying S-layer characteristics likely play a crucial role in environmental adaptation. While the disordered region varies significantly in sequence, the length of the coiled-coil stalk and the MM-binding segment remains highly conserved among PS2 homologs across species (S5 and S6 Figs). This is in line with the fact that the underlying cell envelope architecture, including the MM, is preserved among different *Corynebacterium* species, necessitating the conservation of the MM anchoring segments in PS2. AlphaFold2 models predict that the MM-binding segment consists of an N-terminal hydrophobic α-helix and a short C-terminal amphipathic α-helix; however, in the MM, these may function as a single continuous helix. The MM-binding segment of PS2 homologs in corynebacteria is consistently approximately 25 amino acid residues long, corresponding to a ~3.75 nm α-helix, which is long enough to traverse a significant length of the ~7 nm thickness of the MM. Notably, this segment includes the last residue of PS2, a phenylalanine (F), which is remarkably conserved across all PS2 homologs (S5 and S6 Figs). While the functional significance of this invariant F residue remains unclear, the conservation of the preceding residues, particularly the penultimate residue, which is typically either a proline (P) or lysine (K), suggests a potential functional role. It is plausible that these terminal residues collectively contribute to the sorting, export, and insertion of PS2 into the MM or help ensure its stable anchoring within the lipid-rich MM.

To explore this hypothesis, we analyzed MM-associated proteins of *C. glutamicum* identified in a previous study [52]. Proteins associated with the inner leaflet of the MM, such as the mycoloyltransferases MytA, MytB, MytC, MytD, and MytF, or those involved in pore formation, such as PorA and PorB, do not possess a phenylalanine as their terminal residue, suggesting that the invariant phenylalanine in PS2 does not represent a general mechanism for targeting proteins to the MM. However, several putative cell-surface proteins with a PS2-like C-terminal hydrophobic anchor preceded by a disordered segment were found to harbor a F, P, or K residue at their C-terminus. Examples include a prenyltransferase/squalene oxidase repeat-containing protein (NCBI: WP_011013715.1) and a metallophosphoesterase family protein (WP_011015494.1) (S8 Fig). Based on this conservation, we identified additional putative MM-associated cell-surface proteins in *C. glutamicum* (S8 Fig), such as an ExeM/NucH family extracellular endonuclease (WP_003854007.1) and a lamin tail domain-containing protein (WP_004567709.1). Interestingly, the targeting of porins PorA, PorH, PorB, and PorC to the MM in *C. glutamicum* has been shown to depend on posttranslational O-mycoloylation, which facilitates their proper localization and integration into the MM [53]. Whether O-mycoloylation also plays a role in PS2 targeting remains an open question and warrants further investigation. We speculate that terminal residues such as F, P, and K may contribute to anchoring cell-surface proteins within the MM by stabilizing interactions with the hydrophobic membrane environment or acting as signals for specific sorting or assembly mechanisms.

## Discussion

In this study, we visualized the *C. glutamicum* cell envelope by imaging FIB-milled cells using cryo-EM and cryo-ET, allowing us to report a detailed model of the corynebacterial cell envelope (Fig 5). Although this cell envelope had previously been visualized in impressive detail using the CEMOVIS method [29,31], our study leverages the latest cryo-ET technology, which overcomes several limitations of CEMOVIS, particularly sample deformation caused by the diamond knife

## Schematic Representation of *C. glutamicum* Cell Surface

**Fig 5. Schematic representation of the *Corynebacterium glutamicum* cell surface.** The model depicts the layers of the *C. glutamicum* cell envelope. The S-layer forms the outermost layer, followed by the MM (mycomembrane). The MM is linked to the arabinogalactan layer (OWZ: Outer Wall Zone), which is covalently linked to peptidoglycan layer (MWZ: Medial Wall zone). Below the MWZ is the IWZ (Inner Wall Zone), which contains the granular layer. The innermost layer is the IM (inner membrane), which encloses the cellular contents.

[54,55]. We report several important characteristics of the cell envelope of *C. glutamicum* using our imaging approach (Fig 1). We observed a layer in the cell envelope situated just below the MM, which we have termed the OWZ. This layer forms a less dense region between the MM and MWZ layers and contains some densities that connect the MM to the MWZ. We propose that this region likely corresponds to AG, which is known to be covalently linked both to PG and the MM [14]. Below the PG layer, we observed the IWZ and GL, in line with previous observations (Figs 1 and 5). Our measurements are largely consistent with previous results [31], except that the IWZ in our data was significantly thinner, measuring 8.9–10.5 nm (in cell envelopes with and without an S-layer (S1 Table)), compared to ~18 nm by Zuber *and colleagues.* This difference may be attributed to strain variation. Moreover, our measurement of MWZ was slightly different because we could resolve the OWZ as a separate layer, which was included into the MWZ measurement in the previous study (~13.4–14.2 nm in cell envelopes with and without an S-layer respectively (S1 Table)) in this study versus ~20.9 nm in the previous study [31]. This assignment of the densities in FIB-milled images provides an ultrastructural framework for the future study of the cell envelopes of corynebacteria and mycobacteria (i.e., members of the family *Mycobacteriaceae*), and our images will serve as a reference point for future inquiries into this topic (Fig 5). In addition to cell envelopes of non-dividing cells, the dividing *C. glutamicum* septum shows two daughter cell envelopes separated by a bilayer, likely containing mycolic acids. Notably, this bilayer was not connected to the MM on the rest of the cell (S1E Fig). This observation is in line with the previous studies showing that at septal junctions, a contiguous PG layer acts as a diffusion barrier for the MM, and during separation of daughter cells, the PG in the septal junctions is displaced, allowing the bilayer at the septum to merge with the rest of the MM [56].

In addition to the overall characterization of the corynebacterial cell surface, we focused on one of the most abundant proteins on this cell surface, which also forms the outermost surface—the S-layer. We structurally characterized the PS2 S-layer of *C. glutamicum* using biochemical purification and cryo-EM structure determination (Figs 2-3). PS2 is an SLP composed entirely of α-helices, an unusual feature among prokaryotic S-layers [2,9]. PS2 possesses a C-terminal

coiled-coil-containing segment that anchors it to the MM. Our analysis showed that the MM-anchoring segment, including the coiled-coil, is conserved across all corynebacterial PS2 proteins, indicating its importance in anchoring the S-layer to the cell. This coiled-coil segment may function as a molecular spacer between hydrophobic MM and the hydrophilic S-layer, potentially creating an enclosed space for increasing the local concentration of certain secreted macromolecules. This arrangement could facilitate specific biochemical reactions near the cell surface, in the same manner as for other prokaryotic S-layers [3,42].

The PS2 SLP has a distinctive mode of attachment to the cell envelope compared to other bacterial and archaeal S-layers. In most archaea, S-layers are directly anchored to the cytoplasmic membrane [2], either via lipid modification of the SLP [42] or through interactions with a secondary protein [3]. In diderm, Gram-negative bacteria such as *C. crescentus*, S-layers attach non-covalently to the O-antigen of the LPS layer covering the OM [57]. In contrast, monoderm bacterial S-layers are typically non-covalently anchored to PG-linked secondary cell wall polymers via SLH domains [58]. Some diderm bacteria that stain Gram-positive, such as *D. radiodurans*, employ a different strategy, where the SLP [13] is lipidated at its N-terminus [16], enabling direct interaction with the OM. *C. glutamicum*, however, employs a fundamentally different anchoring mechanism: the PS2 S-layer is secured by the insertion of its C-terminal hydrophobic helix into the MM, a unique feature among structurally characterized bacterial S-layers.

We have further shown that the PS2 S-layer is formed by several acidic amino acid residues (with an estimated isoelectric point of 4.25), making this an extremely hydrophilic surface modification on *C. glutamicum* cells. Given the hydrophobic nature of membranes, including the MM [59], the presence of the S-layer likely modifies the local charge at distinct locations on the cell surface that possess an S-layer coating, thereby significantly influencing the bacterium's interactions with its environment. Although the function of the *C. glutamicum* S-layer is still an open question, we propose that the patchy S-layer observed on *C. glutamicum* modulates the surface properties of the cell envelope in a manner similar to extended surface appendage like pili [5] or adhesins [60]. We also observed that S-layer coverage appeared to increase when *C. glutamicum* cells were grown on solid media (S2A-B Fig). This suggests that the S-layer may facilitate bacterial grow in colonies or surface-attached biofilm communities, as observed in *Clostridium difficile* and *Tannerella forsythia* [61–63]. The large pores in the S-layer, particularly the 27 and 81 Å pores, in conjunction with its patchy cell surface coverage suggest that the S-layer does not function primarily as a protective barrier against invading molecules or phages. It is notable that in other S-layers with large pores in the lattice, such as the *D. radiodurans* S-layer made of HPI protein [13,16], a similarly patchy coverage of the S-layer lattice was observed, supporting this idea. Interestingly, despite its large pores, the PS2 S-layer has been shown to increase cell resistance to lysozyme [38,64]. Although lysozyme is much smaller than the pore sizes, the S-layer may biochemically sequester such undesirable molecules.

Our bioinformatic analysis of the PS2 protein sheds light on the evolution of corynebacterial cell surfaces. Phylogenetic analysis reveals a clear distinction between the PS2-lacking, diphtheria toxin-associated primary pathogens and the PS2-containing environmental and mostly opportunistic pathogens. This suggests that primary pathogens lost the energetically costly S-layer as they adopted a pathogenic lifestyle. The conservation of the C-terminal coiled-coil and anchoring segments, compared to the relative divergence of other PS2 residues involved in lattice formation, further implies that while the S-layer has diverged within corynebacteria, the MM-anchoring mechanism has likely remained the same.

## Materials and methods

### *C. glutamicum* microbiology

*Corynebacterium glutamicum* 541 (ATCC-13058) cells were cultured in liquid BE (beef extract-peptone) media at 30 °C with shaking at 180 rpm (revolutions per minute) until an $OD_{600}$ (optical density measured at 600 nm wavelength of light) of 1.2 was reached. For culturing on solid media, cells were inoculated on BE agar plates (1.5% w/v agar) and incubated at 30 °C for 18 h.

PLOS Biology

## Purification of the PS2 S-layer

Purification of the PS2 S-layer was performed by adapting a previously used protocol [9,16]. Briefly, *C. glutamicum* cells were grown until late-log phase and harvested by centrifugation at 4,000 rcf (relative centrifugal force) at 4 °C for 20 min. The pellet of a 2 liter-culture was resuspended in 100 ml lysis buffer (50 mM HEPES(4-(2-hydroxyethyl)-1-piperazineethanesulfonic acid)/NaOH pH = 7.5, 150 mM NaCl, 1 mM $MgCl_2$, 2 mM $CaCl_2$, 50 μg/mL DNaseI, 0.2 mM TCEP (tris(2-carboxyethyl)phosphine), 2 cOmplete Protease Inhibitor tablets (Roche)). Next, SDS was added to the cell suspension to a final concentration of 2% (w/v). The resulting cell suspension was incubated for 70 min at room temperature (21 °C) and sonicated at 40% amplitude for 5 s in an ice bath. Cell debris was spun down at 4,000 rcf for 15 min at 20 °C and the supernatant was spun down once more at 35,000 rcf for 25 min at 20 °C. The pellet was washed with 1 ml wash buffer (50 mM HEPES/NaOH pH = 7.5, 150 mM NaCl, 1 mM $MgCl_2$, 2 mM $CaCl_2$) and spun down at 20,000 rcf for 15 min at 4 °C. The final pellet was resuspended in 120 μl of wash buffer for cryo-EM grid preparation. For EDTA/EGTA treatment, the purified S-layer was incubated with either 10 mM EDTA (Ethylenediaminetetraacetic acid) or 10 mM EGTA (Ethylene glycol bis(2-aminoethyl ether)-N,N,N''N'-tetraacetic acid) for 2 h at room temperature prior to cryo-EM grid preparation.

## Cryo-EM grid preparation

Cryo-EM grids were prepared by adapting a previously established workflow for S-layers [9,16]. Briefly, 3.5 μl of resuspended sample was applied to a freshly glow discharged Quantifoil R2/2 Cu/Rh 200 mesh grid and plunge-frozen into liquid ethane maintained at −178 °C, using a Vitrobot Mark IV after a wait time of 40 s, with −5 blot force and 2.5 s blot time. For grids containing single cells for cryoEM imaging: cells were grown in BE media until late-log phase and 3.5 μl of cell suspension was applied to a freshly glow discharged Quantifoil R3.5/1 Cu/Rh 200 mesh grid and plunge-frozen using a Vitrobot Mark IV after a wait time of 40 s with −8 blot force and 2.5 s blot time. For cryo-FIB milling: cells were grown in BE media until late-log phase and spun down and resuspended in BE media supplemented with 2.5% glycerol, to an $OD_{600}$ of 30. The resulting concentrated sample was applied (3.5 μl) to Quantifoil R1/4 $SiO_2$/Au 200 mesh grids and plunge-frozen into liquid ethane using a Vitrobot Mark IV after wait time of 3 min with −7 blot force and 13 s of blot time.

## Cryo-FIB milling

Cryo-FIB milling of the *C. glutamicum* cells was performed using an Aquilos2 dual beam FIB-SEM (FIB-scanning electron microscope, ThermoFisher). After loading the frozen specimen into the microscope, the grids were sputter coated with metallic platinum (0.1 mbar; 30 mA, 1 V, 30 s), then mapped using the MAPs software (v. 3.27; ThermoFisher). A coating of organic platinum was then applied via the gas injection system in line with previous protocols [65] for 20 s, followed by a final sputter-coat, using the same settings as applied previously. AutoTEM (v. 2.4.3; ThermoFisher) was set to rough mill the lamellae in three steps with currents of 0.5 nA, 0.3 nA, and 0.1 nA and with milling angle of 10°. Lamellae polishing was performed at 50 pA and then finally at 30 pA current with a 0.5° overtilt to produce lamellae with a final 100–200 nm thickness.

## Cryo-EM single-particle data collection and processing

Cryo-EM data collection was performed as described previously [9] with a Titan Krios G4 transmission electron microscope equipped with cFEG, Falcon 4i direct electron detector and a Selectrix X energy filter used with slit width of 10 eV (ThermoFisher ) at a nominal 105,000× magnification, resulting in a calibrated pixel size of 0.611 Å in super-resolution counting mode with 30° stage tilt. The EPU software (v. 3.7.1; ThermoFisher) was used to record movies in EER format with a total electron dose of 50 e$^-$/Å$^2$ with defoci varying between −1.6 and −2.2 μm (S3 Table).

Cryo-EM data processing was performed using CryoSPARC v4.5.3 [66]. Movies were pre-processed using patch-motion correction and patch CTF estimation. Three hundred particles were manually picked and classified in

2D. With the generated 2D classes, reference-based particle picking was performed. Picked particles were inspected and filtered based on their normalized cross-correlation and power threshold values. Selected particles were extracted with box size of 512 pixel$^2$ and Fourier cropped to 256 pixel$^2$. Particles were further classified in 2D and those particles belonging to poorly averaged classes were discarded. From the best-looking classes, an ab initio volume was produced by using C6 symmetry. From the cleaned particle set, two volumes were produced with heterogenous refinement with C6 symmetry imposed. Particles belonging to the volume with higher resolution were further classified in 2D and poorly refined particles were discarded. The final particle subset was used for a homogenous refinement job which produced a cryo-EM map at 3.47 Å. This particle subset (50,974 particles) was re-extracted with 720 pixel$^2$ box size (Fourier cropped to 360 pixel$^2$) and local CTF refinement was performed. These steps improved the resolution of the volume to 3.35 Å with homogenous refinement. Subsequent non-uniform refinement resulted in a volume at 3.1 Å resolution. The half maps of these jobs together with the calculated FSC mask was used as input for a RELION 5.0 postprocessing, which confirmed the resolution as 3.1 Å, and the phase randomized FSC curve showed no correlation at high resolutions (S3B Fig) [67,68].

Two-dimensional projection images of FIB-milled lamellae were collected with Titan Krios G3i transmission electron microscope (Thermo Fisher) equipped with K3 direct electron detector (Gatan) and Quantum energy filter (slit width 20 eV). Prior to data collection, the angle of lamella with respect to the stage was detected (either +10 or −10) and the specimen stage was tilted to bring the lamella perpendicular to beam path. Data collection was performed manually with the SerialEM software [69] at a nominal 42,000× magnification, resulting in a calibrated pixel size of 1.064 Å in super-resolution counting mode with 66 e$^-$/Å$^2$ total electron dose and defoci varying between −3 and −5 μm (S2 Table). Pre-processing of the movies was performed with patch-motion correction using CryoSPARC v4.5.3 [66]. The distance measurements were performed on the 2D projection images by generating density profiles, which were calculated on rectangular selections (20 nm × 60 nm) by using Fiji [70].

## PS2 atomic model building and refinement

The final volume of the PS2 hexamer was sharpened in CryoSPARC with B-factor of −56 prior to model building. As an additional test, the volume was also sharpened with DeepEMhancer [71] for visual inspection (not used for refinement). The exact sequence of the *ps2* gene from *C. glutamicum* 541 ATCC-13058 was obtained by whole genome sequencing of extracted genomic DNA (shown in S5-S6 Figs). A partial/truncated model was automatically generated using the sharpened map by running ModelAngelo [72]. Combining the output from ModelAngelo with the AlphaFold2 prediction, a composite starting model was generated [73]. Following initial rigid-body fitting, the rest of the model was inspected and manually corrected in Coot 0.9.8 [74]. The sequence register was confirmed based on side chains in the best-resolved regions.

The model building was initially completed for a monomer manually using ISOLDE 1.8 and Coot followed by real space refinement performed in PHENIX 1.20.1 [74–77]. Next, four interacting monomers (in the hexameric and trimeric interfaces) were added to the model and refined together to ensure proper refinement of the inter-subunit interfaces. In the final rounds of real space refinement, a hexameric model was refined with non-crystallographic symmetry constraints by PHENIX real-space refinement. The atomic model validation was performed within the PHENIX suite (S3 Table). ChimeraX was used for data visualization [78].

## Cryo-ET data collection and processing

Cryo-ET data was collected as previously described [57]. Briefly, data collection was performed with Titan Krios G3i transmission electron microscope (Thermo Fisher) equipped with a K3 direct electron detector (Gatan) and Quantum energy filter (slit width 20 eV). Prior to data collection, the angle of lamellae with respect to the stage was detected and the stage was tilted to have the lamellae perpendicular to beam path. Tilt series were then acquired with the SerialEM software [69]

using a dose-symmetric scheme [79] at a nominal 42,000× magnification (calibrated pixel size of 2.13 Å) in counting mode with defoci between −5 and −8 μm, ±60° oscillation and 2° tilt increment with a total dose of 122 e⁻/Å² (S2 Table).

Movies of each tilt image were aligned with MotionCor2 [80]. Tilt series alignment and tomogram generation were performed with AreTomo [81]. Tomograms were denoised with CryoCare [82]. Further visual inspection was performed within the IMOD package [83]. The overlay of the S-layer atomic model in tomograms was done using ArtiaX in ChimeraX [78,84].

**Bioinformatic analysis.**

All sequence similarity searches were performed using the BLAST web server at NCBI [45] and HHsearch in the MPI Bioinformatics Toolkit [46,47]. The searches were seeded with the PS2 protein sequence from *C. glutamicum* ATCC 13058. To assess the prevalence of PS2 in species within the *Corynebacterium* genus, we downloaded a total of 2,315 *Corynebacterium* proteomes from the RefSeq genomes database [85], filtering out duplicate proteomes from identical isolates to retain 2,256 unique proteomes for further analysis. These proteomes were pooled, and a custom BLAST database was built using *makeblastdb* with default settings. We then searched for the PS2 protein across *Corynebacterium* species by running BLAST with default settings, using the PS2 protein sequence from *C. glutamicum* ATCC 13058 as the query against our custom *Corynebacterium* database. Signal peptides were predicted using SignalP 6.0 [86], and multiple sequence alignments were computed using PROMALS3D [87]. The phylogenetic trees of *Corynebacterium* species and *C. glutamicum* isolates, shown in Fig 4, were generated using the de novo workflow (*de_novo_wf*) of the GTDB-Tk 2.4.0 software toolkit (GTDB) [26,51] with default settings. For the tree generation, uncultured and undetermined species were omitted. The phylogenetic tree of *Corynebacterium* species contains 1,585 species, including three outgroup species used for rooting the tree (*Dietzia maris* IMV 195T; NCBI GCF_014144855.1, *Lawsonella clevelandensis* X1036; GCF_001293125.1, and *Mycobacterium tuberculosis* H37Rv; GCF_000195955.2). The tree of *C. glutamicum* isolates contains 73 isolates and one outgroup species used for rooting (*C. crudilactis* JZ16; GCF_001643015.1). The trees were visualized using iTOL [88]. Structural models of PS2 from representative species were predicted using AlphaFold v2.3.2 [73].

## Supporting information

**S1 Fig. Cryo-FIB milling of *Corynebacterium glutamicum* cells. (A)** Cryo-EM images of *C. glutamicum* cells deposited on grids without FIB milling. The S-layer decorates the *C. glutamicum* cell envelope in a patchy manner. S-layer (surface layer), MM (mycomembrane), and IM (inner membrane) are marked. **(B)** FIB-milling of *C. glutamicum* cells. Grids made for FIB-milling contained clumps of *C. glutamicum* cells, providing several suitable areas for milling. After milling, lamellae with a 150–200 nm thickness were retained for cryo-ET investigations. Each lamella contained multiple cells suitable for imaging. Although vitreous ice was observed in most lamellae, the edges of some lamellae showed signs of crystalline ice formation. **(C)** Slices from a tomogram of a dividing *C. glutamicum* cell (S1 Movie). The slices were bandpass-filtered to enhance contrast. The first slice was used as a reference point to calculate *Z*-values (marked in nm). Red inset shows a zoom of the site of division. **(D)** 2D projection images of FIB-milled *C. glutamicum* cells. The 2D projection images show high-contrast details of the cell envelope. The S-layer, MM, and IM are marked. The images were Gaussian-filtered to enhance contrast. **(E)** Septum of a dividing *C. glutamicum* cell. Ten 0.85 nm thick slices of the tomogram were averaged and bandpass-filtered to boost contrast. Zoomed view of the septum is shown on the right.
(TIF)

**S2 Fig. Cryo-EM of *Corynebacterium glutamicum* cells.** Cryo-EM images of *C. glutamicum* cells grown **(A)** on solid media, **(B)** in liquid media deposited on grids. S-layer coverage appeared to be increased in cells grown on solid media when compared to cells grown in liquid media, although the top and bottom parts of the cells are not interpretable in 2D images (and absent in milled lamellae). S-layer (surface layer) and MM (mycomembrane) are marked.
(TIF)

**S3 Fig. Cryo-EM analysis of the PS2 S-layer. (A)** Local resolution estimates plotted onto the cryo-EM map of the PS2 S-layer, colored based on the local resolution values shown in the color key (bottom right). **(B)** Fourier Shell Correlation (FSC) resolution estimation of the cryo-EM map. A gold-standard FSC estimates the global resolution of the map as 3.1 Å. The data underlying this plot can be found in S2 Data. **(C)** PS2 atomic model shown as a ribbon diagram in two orthogonal views, colored in a rainbow gradient with α-helices α1-α8 labeled. **(D)** Representative density fits of the PS2 model. **(E)** The density related to C-terminus of PS2 is disordered and could not be used for model building. The shown map has been Gaussian-filtered with a width of 0.9 Å to illustrate the unmodeled density at the tip of coiled-coil segment. (TIF)

**S4 Fig. Features of the PS2 S-layer lattice. (A)** The cryo-EM map of the PS2 S-layer is shown in color to illustrate the long-range lattice connections. Each color represents a different monomer of PS2, labeled from 'a' to 'f', repeated along the lattice. **(B)** Electrostatic potential map of the PS2 hexamer. **(C)** Hydrophobicity map of the PS2 hexamer. **(D–F)** Putative densities possibly corresponding to cations and **(G)** SDS detergent molecules are shown, with the respective sigma values of the maps shown in the bottom right. The potential densities are denoted with an "*", and the surrounding residues are also labeled. **(H)** The coiled-coil segment (residues 405–445) is shown in side-view (left) and bottom-view (right). **(I)** Purified PS2 S-layer sheets incubated with EDTA (middle) and EGTA (right) show no discernible differences from native S-layers (left). (TIF)

**S5 Fig. Multiple sequence alignment of PS2 from representative *Corynebacterium glutamicum* isolates.** The alignment of PS2 sequences from the following strains is shown: ATCC 13058 (this study), ATCC 14751 (NCBI: AAS20296.1), CICC10065 (WP_040967778.1), ATCC 19240 (AAS20313.1), 22243 (AAS20307.1), and ATCC 17966 (AAS20302.1). Signal peptides in the sequences are highlighted in green, α-helices in yellow-green, and the MM-binding segment in light red. Coiled-coil (C) and intrinsically disordered (D) regions are indicated. Secondary structure was assigned based on our experimental model of *C. glutamicum* PS2 and AlphaFold2 models. The alignment was computed using PROMALS3D. In the consensus line, conserved amino acids are represented by bold and uppercase letters, with the following annotations: aliphatic (I, V, L): l; aromatic (Y, H, W, F): @; hydrophobic (W, F, Y, M, L, I, V, A, C, T, H): h; alcohol (S, T): o; polar residues (D, E, H, K, N, Q, R, S, T): p; tiny (A, G, C, S): t; small (A, G, C, S, V, N, D, T, P): s; bulky residues (E, F, I, K, L, M, Q, R, W, Y): b; positively charged (K, R, H): +; negatively charged (D, E): −; charged (D, E, K, R, H): c. (TIF)

**S6 Fig. Multiple sequence alignment of PS2 from representative *Corynebacterium* species.** The alignment of PS2 sequences from the following species is shown: *C. glutamicum* ATCC 13058 (this study), *C. auris* DSM 44122 (WP_290341829.1), *C. glaucum* DSM 30827 (WP_095660674.1), *C. minutissimum* NCTC10289 (WP_115020885.1), and *C. aurimucosum* UMB1300 (WP_102234280.1). The sequences are annotated as in S5 Fig. (TIF)

**S7 Fig. AlphaFold2-predicted monomeric structures of PS2 from representative species.** In all structures, α-helices are colored red. In the superimposed structures, one of the chains is shown in light red. (TIF)

**S8 Fig. Putative MM-binding cell surface proteins in *Corynebacterium glutamicum*.** Predicted signal peptides are colored green, intrinsically disordered regions are colored blue, and the putative MM-binding segments are colored red. (TIF)

**S1 Table. Dimensions of cell envelope layers of *Corynebacterium glutamicum*.** Measurements were performed using the line profiles averaged over 20 nm along the cell envelope. Apart from the membranes, measurements were performed by measuring the peak-to-peak distances in the profiles. For reference, the peak-to-peak measurements for

the membranes are as follows: MM without S-layer 3.1 ± 0.2 nm, MM with S-layer 3.0 ± 0.3 nm, IM without S-layer 3.8 ± 0.3 nm, and IM with S-layer 3.8 ± 0.5 nm. The membrane dimensions were measured directly from the micrographs. For each data point shown, six different measurements were performed, and standard deviations were calculated. The data underlying this table can be found in S1 Data.
(PDF)

**S2 Table. Parameters for cryo-EM and cryo-ET data collection from FIB-milled cells.**
(PDF)

**S3 Table. Cryo-EM data collection, image processing, and model refinement statistics of the purified PS2 S-layer.**
(PDF)

**S1 Movie. Cryo-ET of a FIB-milled *Corynebacterium glutamicum* cell.** A cryo-electron tomogram of FIB-milled *C. glutamicum* cell is shown, with the mycomembrane and inner membrane marked.
(MP4)

**S2 Movie. Cryo-ET of a FIB-milled *Corynebacterium glutamicum* cell.** A cryo-electron tomogram of different FIB-milled *C. glutamicum* cell is shown, with the mycomembrane, inner membrane, and the S-layer marked.
(MP4)

**S3 Movie. Cryo-EM structure of the PS2 S-layer.** The cryo-EM map and atomic structure of the *Corynebacterium glutamicum* PS2 S-layer is shown. Different views of the PS2 monomer and the S-layer lattice are shown with text annotations.
(MP4)

**S1 Data. Cell_envelope_measurements.csv.** Cell envelope measurements presented in S1 Table are provided in CSV format.
(CSV)

**S2 Data. FSC_curve.csv.** FSC values for the plot in S3B Fig are provided in CSV format.
(CSV)

**S3 Data. Sequence_list.csv.** A list of sequences of the PS2 protein are provided in CSV format.
(CSV)

**S4 Data. Fig4A.tree.** The tree file for the phylogenetic tree is shown in Fig 4A.
(TREE)

**S5 Data. Fig4B.tree.** The tree file for the phylogenetic tree is shown in Fig 4B.
(TREE)

## Acknowledgments

The authors would like to thank Abul Tarafder, Ido Caspy, Kenny Jungfer, and Jan Böhning for critically reading this manuscript. We acknowledge the MRC LMB electron microscopy facility for help with sample preparation and data collection and the MRC LMB mass spectrometry facility for assistance with mass spectrometry analysis of the sample. V.A. would like to thank Andrei Lupas for continued support.

## Author contributions

**Conceptualization:** Buse Isbilir, Vikram Alva, Tanmay A. M. Bharat.

**Data curation:** Buse Isbilir, Vikram Alva, Tanmay A. M. Bharat.

**Formal analysis:** Buse Isbilir, Vikram Alva, Tanmay A. M. Bharat.

**Funding acquisition:** Vikram Alva, Tanmay A. M. Bharat.

**Investigation:** Buse Isbilir, Anna Yeates, Vikram Alva, Tanmay A. M. Bharat.

**Methodology:** Buse Isbilir, Anna Yeates, Vikram Alva, Tanmay A. M. Bharat.

**Project administration:** Vikram Alva, Tanmay A. M. Bharat.

**Resources:** Vikram Alva, Tanmay A. M. Bharat.

**Supervision:** Tanmay A. M. Bharat.

**Validation:** Buse Isbilir, Vikram Alva, Tanmay A. M. Bharat.

**Visualization:** Buse Isbilir, Anna Yeates, Vikram Alva, Tanmay A. M. Bharat.

**Writing – original draft:** Buse Isbilir, Vikram Alva, Tanmay A. M. Bharat.

**Writing – review & editing:** Buse Isbilir, Anna Yeates, Vikram Alva, Tanmay A. M. Bharat.

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
