## [Editor Report · Decision Letter 0]

24 Feb 2025

Dear Dr Bharat,

Thank you for submitting your manuscript entitled "Mapping the ultrastructural topology of the corynebacterial cell surface" for consideration as a Short Reports by PLOS Biology.

Your manuscript and the revision of it has now been evaluated by the PLOS Biology editorial staff, as well as by an academic editor with relevant expertise, and I am writing to let you know that we would like to send your submission out for external peer review. While we appreciate the comments from the previous reviewers and your revision of them, we would like for an expert on mycobacterial cell envelope to peer-review to see if they perceive that the impact could extend beyond Corynebacteria.

However, before we can send your manuscript to an additional reviewer, we need you to complete your submission by providing the metadata that is required for full assessment. To this end, please login to Editorial Manager where you will find the paper in the 'Submissions Needing Revisions' folder on your homepage. Please click 'Revise Submission' from the Action Links and complete all additional questions in the submission questionnaire.

Once your full submission is complete, your paper will undergo a series of checks in preparation for peer review. After your manuscript has passed the checks it will be sent out for review. To provide the metadata for your submission, please Login to Editorial Manager (https://www.editorialmanager.com/pbiology) within two working days, i.e. by Feb 26 2025 11:59PM.

Kind regards,

Melissa

Melissa Vazquez Hernandez, Ph.D.

Associate Editor

PLOS Biology

---

## [Decision Letter · Decision Letter 1]

11 Mar 2025

Dear Dr Bharat,

Thank you for your patience while we considered your revised manuscript "Mapping the ultrastructural topology of the corynebacterial cell surface" for publication as a Short Reports at PLOS Biology. This revised version of your manuscript has been evaluated by the PLOS Biology editors, the Academic Editor and an additional reviewer.

Based on the additional review, we are likely to accept this manuscript for publication, provided you satisfactorily address the remaining points raised by the reviewer. Specifically, please modify the text as advised by the reviewer to make it clear that the S-layer is Corynebacteria-specific and to acknowledge the open questions about S-layer function. Please also make sure to address the following data and other policy-related requests.

Please supply the numerical values either in the a supplementary file or as a permanent DOI’d deposition for the following figures:

Figure S3B and table S1

b) Please cite the location of the data clearly in all relevant main and supplementary Figure legends, e.g. “The data underlying this Figure can be found in S1 Data” or “The data underlying this Figure can be found in https://doi.org/10.5281/zenodo.XXXXX”

c) Please provide the tree files for the phylogenetic trees in Figures 4AB

d) Because several findings are related to microscopy images, we would like to encourage you to upload any other microscopy pictures to either Figshare or Zenodo, so the findings have more support

e) Please ensure that your Data Statement in the submission system accurately describes where your data can be found and is in final format, as it will be published as written there.

f) Per journal policy, if you have generated any custom code during the course of this investigation, please make it available without restrictions upon publication. Please ensure that the code is sufficiently well documented and reusable, and that your Data Statement in the Editorial Manager submission system accurately describes where your code can be found.

We expect to receive your revised manuscript within two weeks.

*Published Peer Review History*

*Press*

Sincerely,

Melissa

Melissa Vazquez Hernandez, Ph.D.

Associate Editor

PLOS Biology

Reviewer #1:

In this manuscript, the authors study the S-layer of Corynebacterium glutamicum. They show that the S-layer is distributed patchily around cells, and that the MM layer is discontinuous in regions where there is an S-layer. They use Cryo-EM to make a structural description of S-layer oligomers . They use phylogenetic analysis to show that homologs of PS2, the S-layer protein, are only found in Corynbacterium genus, and are not found in most Coryne species.

The lack of conservation limits the significance of this work: S-layer is not found in other genus of the Mycobacteriaceae family. Is only found in some species of Corynebacterium. No data are presented that indicate a physiological role for the S-layer. In the discussion the authors mention that the S-layer contributes to lysozyme resistance - this is apparently the only information available that speaks to the significance of this structure - maybe this should be mentioned in the introduction?

In the abstract, the first paragraph implies that this work is relevant to mycobacterial pathogens. However, this work is focused on the S-layer, and the authors themselves show that the S-layer protein is not found in any Mycobacteria. The abstract and Introduction (eg, likes 94-97) should be re-written so as not to mis-lead readers about the impact of this work. This work does not provide insights into the cell envelope of mycobacteria. I suggest, in the Abstract and Introduction, focusing more on the general roles of S-layers, which should involve a literature review of other organisms that contain S-layers, not of the mycobacteria.

Line 26 - this is inaccurate, many ultrastructural details of the mycobacterial cell envelope are well described. Please be more specific about what is unknown and what you are addressing.

Introduction can be edited: background information at the introductory undergraduate level is not required.

Fig S1 - the insets in panel A are too small to see anything. Since the insets are where the important information is, maybe make them bigger and make the zoom-out pics smaller?

Fig 4 line 323 - please explain briefly how this phylogenetic tree was built in the results section - what models are used to infer relationships? What data are used? Please spell out what GTDB stands for in lines 538 and 846.

Lines 324-325 - can you please explain the logic behind this conclusion more? Why is your assumption that the common ancestor contained an S-layer? Instead of that S-layer genes could be horizontally transferred?

Lines 486-490 - this conclusion is unsubstantiated. You are trying to make conclusions about the composition of the membrane from the sequence of proteins that insert into that membrane? This doesn't make sense to me - if there is some way to make such inferences, please provide citations. There are already direct data about the composition of mycomembranes in these species. Also, your paper is not about the mycomembrane, so this seems like a strange final statement.

---

## [Editor Report · Decision Letter 2]

25 Mar 2025

Dear Tanmay,

Thank you for the submission of your revised Short Reports "Mapping the ultrastructural topology of the corynebacterial cell surface" for publication in PLOS Biology. On behalf of my colleagues and the Academic Editor, Erin Danielle Goley, I am pleased to say that we can in principle accept your manuscript for publication, provided you address any remaining formatting and reporting issues. These will be detailed in an email you should receive within 2-3 business days from our colleagues in the journal operations team; no action is required from you until then. Please note that we will not be able to formally accept your manuscript and schedule it for publication until you have completed any requested changes.

PRESS

Sincerely, 

Melissa

Melissa Vazquez Hernandez, Ph.D., Ph.D.

Associate Editor

PLOS Biology
